# Correlation of Kidney Size on Computed Tomography with GFR, Creatinine and HbA1C for an Accurate Diagnosis of Patients with Diabetes and/or Chronic Kidney Disease

**DOI:** 10.3390/diagnostics11050789

**Published:** 2021-04-27

**Authors:** Nashaat Ghaith, Bassem Malaeb, Rasha Itani, Mohammed Alnafea, Achraf Al Faraj

**Affiliations:** 1Department of Radiologic Sciences, Faculty of Health Sciences, American University of Science and Technology (AUST), Beirut 1100, Lebanon; nashaat_gh12@hotmail.com (N.G.); rasha_itani@outlook.com (R.I.); 2Department of Radiology, Saint George University Hospital and Research Center, Beirut 1100, Lebanon; 3Department of Radiology, Ain Wazein Medical Village, Chouf 5841, Lebanon; bnmalaeb@gmail.com; 4Department of Radiological Sciences, College of Applied Medical Sciences, King Saud University, Riyadh 11433, Saudi Arabia; alnafea@ksu.edu.sa

**Keywords:** chronic kidney disease, diabetes, computed tomography, glomerular filtration rate, HbA1c, kidney volume, albumin creatinine rate

## Abstract

Diabetes is considered one of the major causes of chronic kidney disease (CKD), affecting renal blood vessels and nerves. Diagnosis of CKD by traditional biochemical serum and blood analyses is insufficient and insensitive, thus requiring the development of a more robust technique. This novel study aims to propose a new method for the accurate diagnosis of CKD, quantification of kidney damage, and its prognosis by physicians by measuring the kidney volume on computed tomography (CT). In total, 251 patients were enrolled in this retrospective study. They were divided into four groups: control, patients having diabetes, patients having CKD, and patients having both diabetes and CKD. Results showed that kidney volume correlated negatively with both GFR and HbA1C on CT images, in addition to decreasing faster in males than females. Moreover, HbA1C was shown to correlate positively with creatinine and negatively with GFR. Finally, GFR was more robust than creatinine when correlated with age. The association between kidney volume with GFR and HbA1c can be used to accurately anticipate kidney volume in established CKD on CT scan, especially in resource-poor settings. Furthermore, HbA1C can serve as a powerful biomarker for studying renal function in diabetic CKD patients as it correlates with creatinine and GFR.

## 1. Introduction

Chronic kidney disease (CKD) is defined as the gradual loss or impairment of kidney function due to damage. CKD is a serious, widespread, global disease that has dramatically increased over the past few years, affecting around 697.5 million people and resulting in 1.2 million deaths around the globe in 2017. It is ranked as the 12th leading cause of death [1]. CKD is manifested by abnormal albumin secretion, assessed by albumin creatinine rate (ACR), and/or abnormal kidney function computed by the glomerular filtration rate (GFR). Kidney disease is considered chronic when these functional impairments are observed over a period exceeding 3 months [2]. According to the 2002 Kidney Disease Outcomes Quality Initiative (KDOQI) guidelines for identifying and treating CKD, it is recommended to classify CKD based primarily on GFR instead of using serum creatinine concentration to assess kidney function [3]. Since the total kidney GFR is equal to the sum of the filtration rates in each of the functioning nephrons, the total GFR can be used as an index of renal mass [4]. CKD is classified into five stages, according to the extent of kidney damage and/or decline in kidney function, the latter reflected by the GFR. Accordingly, CKD is classified as follows: stage 1 involves kidney damage with normal GFR (>90 mL/min/1.73 m^2^); stage 2 reflects kidney damage with mild decrease in GFR (≈60–89 mL/min/1.73 m^2^); stages 3a and 3b encompassing moderate decrease in GFR (≈45–59 mL/min/1.73 m^2^ for 3a and ≈30–44 mL/min/1.73 m^2^ for 3b); stage 4 includes severe decrease in GFR (15–29 mL/min/1.73 m^2^); and stage 5 involves kidney failure with GFR (<15 mL/min/1.73 m^2^) [5]. Usually, patients with end-stage CKD will require dialysis [6].

In addition to laboratory testing, medical imaging techniques have been proposed to assess renal functional and morphological abnormalities [7]. In patients with advanced CKD, ultrasound (US), computed tomography (CT) and magnetic resonance imaging (MRI) can play an important role in assessing morphological changes in the kidney for diagnosis and staging, in addition to directing treatment plan and therapy [8]. US is considered as the first-line imaging technique in cases of patients at risk of kidney damage and/or suffering from kidney disease as it is a simple, cost-effective and non-invasive imaging modality [9]. Conventional US allows the measurement of renal diameter and cortical thickness in various planes, as well as assessing renal echogenicity and examining the urinary and collecting systems. It aids in evaluating the extent of kidney parenchymal damage and estimating the necessity of kidney biopsy [10]. However, renal volume measurement using US is challenging. Due to the complex morphology of the kidney, renal volume remains the most sensitive criterion for detection of renal abnormalities and can represent renal mass more accurately than renal length [11]. Furthermore, US is greatly considered as a user/operator-dependent modality, having various inter- and intra-operator result variabilities, depending on the operator’s skill, measurement methods, interpretation and equipment used. Therefore, CT can be used to precisely measure renal size and renal volume as it is widely used for the preoperative assessment of renal anatomy and can be associated with renal function in contrast-enhanced scans. In the two-dimensional (2D) plane, linear renal dimensions can be measured and renal length can be calculated from axial slices by multiplying the slice thickness by the number of slices between the superior and inferior poles of the kidney. The lateral kidney diameter is measured from the lateral border of the kidney to the renal sinus [12]. However, these 2D measurements are strenuous and time-consuming; additionally, specialized 3D software is not commonly available in imaging institutions. This advanced software is mostly used in comparative studies [13]. Therefore, an easy ellipsoid method was suggested for estimating renal volume, which slightly underestimated the 3D volumes [12]. The only limitation for such an evaluation was the use of nephrotoxic contrast media, a contraindication in cases of renal failure, in contrast-enhanced studies [14]. MRI may be used to accurately calculate kidney volume using multiple consecutive slices over the entire kidney and disc summation methods. In addition to this, a study regarding diffusion tensor imaging (DTI) of the renal cortex in diabetic patients has shown that renal fractional anisotropy (FA) and renal cortex apparent diffusion coefficient (ADC) may help in differentiating diabetic kidneys from volunteers. DTI also helps in predicting the presence of macro albuminuria in diabetic patients, correlated with some of the urinary and serum biomarkers of diabetes [15]. However, similar to a CT scan, the accuracy and feasibility is related to gadolinium-enhanced studies, contraindicated in patients with renal failure.

CKD has various risk factors that play a significant role in disease development and advancement (i.e., natural, aetiologic, environmental, occupational, genetic, lifestyle-related, etc.). Additionally, CKD may also occur as a complication of various other diseases. Diabetes mellitus is considered one of the leading causes of CKD; accordingly, glucose control plays an important role in controlling diabetic nephropathy. Based on this, assessing the time-averaged mean of glycemia, through monitoring the glycosylated hemoglobin (HbA1C) level, is the main criterion for maintaining normal glycemia levels and decreasing the risk of diabetes-associated complications [16]. The 2012 update in the KDOQI guidelines suggested a 7% level of HbA1C to prevent and/or delay the various complications associated with diabetes, such as kidney disease [17]. However, several conducted studies and trials proved that using HbA1C to evaluate glucose control does not reflect the variation in glucose or the risks associated with prolonged abnormal glucose levels [18]. Additionally, tight glucose control and the ability to maintain glucose at its normal ranges (i.e., fasting: 4 to 6 mmol/L and less than 7.8 mmol/L after two hours following a meal) is not only challenging but also not recommended for diabetic patients [19]. Therefore, the frequency of various micro- and macro-vascular complications associated with diabetes is growing rapidly, out of which diabetic kidney disease (DKD) and CKD are the most frequent [20]. The combination of diabetes and hypertension may gradually deteriorate albuminuria and cause a decline in GFR levels, thus leading to CKD [21]. Diabetes causes renal worsening through the production of unnecessary additional reactive oxygen species (ROS) produced by the kidney’s mitochondria as a result of a hyperglycemic state [22]. Studies have shown that abdominal obesity, with waist circumference greater than 102 cm in men and 88 cm in women, can increase the risk of developing CKD bi-fold. It is a mutable risk factor, along with high sodium and potassium intake and absorption, that, in turn, leads to increased microalbuminuria and reduced kidney function [23].

While CKD may be asymptomatic at its early stages, it was reported that it might develop various symptoms such as anemia, damage to glomerular capillary walls, gastrointestinal disturbances, itch and cramps, hematuria, dyspnea, fatigue, poor appetite, hypertension and peripheral edema at its end-stages. CKD may also cause a series of fatal complications, including bone diseases, metabolic disorders, cancer, as well as cardiovascular diseases [24].

This study aims to facilitate the diagnosis of CKD at its early stages and monitor its progression by assessing kidney size on CT scan. It also aims to uncover a correlation between HbA1C and kidney volume in diabetic patients having CKD for better disease management and prevention of complications using CT, especially when the results are reinforced by clinical and laboratory data.

## 2. Materials and Methods

This retrospective study was carried out over a period of 6 months, from 1 September 2020 to 28 February 2021. Data were collected from Ain Wazein Medical Village according to the International Classification of the Diseases Tenth Version (ICD 10), a system used by researchers and physicians to code and classify symptoms, diagnosis and diseases for CKD. It involves specific codes for diabetes and diabetic kidney disease (DKD), in addition to patients’ ID, age, gender, GFR, HbA1C and CT scan images. Patients having previous CT scan and laboratory test results in the hospital information system were included in the study. It is important to note that patients with incomplete data, acute kidney injury/failure, renal malignancy, obstruction, congenital anomalies (i.e., renal hypodysplasia or agenesis, multicystic dysplastic kidney, hydronephrosis, duplex kidney or duplicated collecting system, etc.) and polycystic kidney disease, in addition to those who underwent unilateral nephrectomy, were excluded from this study.

A total of 251 participants were included in this IRB-approved study (CRU263). They were divided into four groups, with the lowest age being 45, since CKD is rare before this age and its presence increases with age, especially after age 65. Group 1 was classified as a control group including healthy patients without CKD or diabetes; Group 2 comprised patients having diabetes without CKD; Group 3 involved patients having CKD without diabetes; and Group 4 involved patients having both CKD and diabetes. Creatinine was measured by the AU480 chemistry analyzer system (Beckman Coulter, Brea, CA, USA) according to the manufacturer’s protocol. GFR was calculated using the Cockcroft–Gault formula and HbA1C was measured by the Tosoh Automated Glycohemoglobin HLC-723G8 Analyzer (Tosoh Corporation, Tokyo, Japan). The identification of CKD and diabetic patients was performed based on GFR and HbA1C results after obtaining previously collected pathological data. CT images were acquired using a 64 multi-detector computed tomography (MDCT) Siemens Somatom Perspective scanner (Siemens AG, Munich, Germany). The CT scan protocol incorporated the following technical parameters: 38.4 mm detector width; 0.6 to 20 mm reconstructed slice width, 80, 110, and 130 kVp ranges; 20 to 345 mA range; 17.5 lp/cm spatial resolution; 50 to 70 cm scan field of view (FOV); and 512 × 512 matrix size. CT images were analyzed independently by two observers with more than 10 years of experience in image data processing and analysis. Inter-observer reproducibility was assessed using intra-class correlation coefficients (ICC). *p* < 0.05 was considered to indicate a significant difference.

Since diabetes affects both kidneys equally, and in order to simplify the data, quantification was performed on the right kidney only. Renal length was measured using pole-to-pole kidney length distance on the coronal CT plane; kidney width was the diameter measured from the renal hilum to the opposite side on the transverse plane, and renal depth (anteroposterior diameter) was the longest distance on an axis perpendicular to the kidney width. Kidney volume, the most accurate measurement of renal size [12], was computed in cubic centimeters using the ellipsoid equation: Volume (cm^3^) = length × width × depth × π/6.

The collected data, including the demographic characteristics, were analyzed using MegaStat 10.1 add-in for Excel (McGraw Hill, New York, NY, USA). To model the relationship between dependent variables such as kidney volume, GFR and creatinine and independent variables such as age and HbA1C, a linear regression model was constructed for each group. All correlations were made at a 99% confidence level and 90% power level. The minimum sample size needed for the whole study was calculated using Cochran’s formula and was found to be 166.

## 3. Results

Kidney volume measured on CT images, the levels of creatinine, GFR and/or HbA1C, along with patients’ age were correlated in the four different patient groups and regression coefficients were calculated. Scatter plot graphs were reported when there was either a positive or a negative correlation.

Healthy patients (Group 1) included 85 patients (46 females and 39 males) with a mean age of 63.85 for females (minimum 48 and maximum 91) and a mean age of 62.36 for males (minimum 47 and maximum 92). A negative correlation was observed between kidney volume and age with Pearson correlation coefficient (*r* = −0.597) in males and (*r* = −0.308) in females (Figure 1).

Moreover, there was a positive correlation between creatinine and age with correlation coefficient (*r* = 0.308) and a negative correlation between GFR and age (*r* = −0.676) (Figure 2).

Diabetic patients (Group 2) included 111 patients (61 females and 50 males) with a mean age of 67.97 for females (minimum 50 and maximum 86) and mean age of 64.78 for males (minimum 47 and maximum 95). There was a very weak negative correlation between HbA1C and age (*r* = −0.031) while a weak positive correlation was seen between kidney volume and HbA1C (*r* = 0.063) (Figure 3).

Chronic kidney disease patients (Group 3) included 24 patients (10 females and 14 males) with a mean age of 72.60 for females (minimum 53 and maximum 88) and mean age of 74.36 for males (minimum 49 and maximum 95). A negative correlation was observed between creatinine and kidney volume (*r* = −0.594) while a positive correlation was noted between GFR and kidney volume (*r* = 0.371) (Figure 4).

Patients having both CKD and diabetes (Group 4) included 31 patients (19 females and 12 males) with a mean age of 75.21 for females (minimum 61 and maximum 86) and a mean age of 72.08 for males (minimum 59 and maximum 86). A negative correlation was found between GFR and HbA1C (*r* = −0.031) and a positive correlation between creatinine and HbA1C (*r* = 0.118) (Figure 5). Interestingly, a positive correlation was observed between kidney volume and HbA1C (*r* = 0.178).

The positive correlation between HbA1C and kidney volume prompted further investigations as it contradicted the study’s expected result. Therefore, Group 4 was divided into two subgroups, controlled diabetes (included 24 patients) versus uncontrolled diabetes (included 7 patients), as controlling diabetes is known as the best way to prevent or slow down kidney damage [25]. The division between the two subgroups was made based on the guidelines issued by the American Diabetes Association (ADA), where it recommended an HbA1c value ≥6.5% for diagnosing diabetes [26]. Furthermore, the duration of diabetes (i.e., how long the patient has been diabetic) was investigated since the maximal incidence of nephropathy was reported as 10 to 20 years following diabetes onset [27]. A positive correlation (r = 0.126) was seen between kidney volume and HbA1C in controlled diabetic CKD patients for less than 20 years. Meanwhile, a stronger negative correlation (r = −0.577) was seen between the same variables but in controlled diabetic CKD patients for 20 years and above (Figure 6).

In uncontrolled diabetic CKD patients (below 20 years), there was a positive correlation between HbA1C and kidney volume (*r* = 0.9333), and a similarly positive correlation between the same variables but in uncontrolled diabetic CKD patients for 20 years and above (*r* = 0.996) (Figure 7).

## 4. Discussion

Advanced glycation end-products, often caused by long-term, persistent hyperglycemia in diabetic patients, can trigger toxic mechanisms that can adversely impact health, especially the kidneys, resulting in diabetic nephropathy. The growth factor that manufactures extracellular proteins can increase significantly, thus leading to disposition of these proteins and, in turn, leading to renal mesangial extension or glomerular sclerosis [28]. In addition, CKD patients may present with initial hyper-filtration, passing by micro- and macroalbuminuria and thus reaching end-stage renal disease [29]. Fortunately, specific medications given during these initial stages can reverse or delay the effect on glomeruli. The initial stages were assessed by early measurement of blood glucose levels and screening for reversible microalbuminuria or nephropathy, especially in recently diagnosed diabetic patients [30].

Hence, this study was conducted with the aim of expanding the screening and diagnosing tools on one hand, and since HbA1C is similarly efficacious as GFR in studying CKD [31] on the other hand. In addition, to the best of our knowledge, no study has been performed to correlate creatinine, GFR, HbA1C and kidney volume measured using CT scan. In this current study, the Cockcroft–Gault formula was employed to estimate GFR (recommended by KDOQI guidelines in CKD patients) in contrary to more recent studies [32] where the modification of diet in renal disease was found to be more precise than the Cockcroft–Gault formula in GFR calculation.

Interestingly, the present study derived several correlations. In healthy patients, the correlation coefficient was greater between age and GFR than between age and creatinine. Consequently, the use of GFR was found to be more powerful when correlating with age, thus supporting the fact that creatinine revealed a weak correlation with age in normal, healthy patients [33].

When correlating between kidney volume and age, males showed a stronger negative coefficient than females. Therefore, based on these results, kidney volume declines faster with age in males than in females. This is not surprising since gender, similar to many other factors, has its own effect on the age-related renal function decline [34]. Piras et al. reported that, during a woman’s life, kidney volume decreases slowly, whereas it progressively decreases in men after the fifth decade [35]. Kidney volume, as stated in various studies, can be a significant prognostic biomarker for risk assessment and disease tracking, as CKD patients usually develop fluid overload associated with systemic inflammation [36]. This volume overload has been linked to a decrease in renal function in advanced CKD patients [37].

This study also showed a very weak, insignificant correlation between HbA1C and age in diabetic patients. However, in patients having both CKD and diabetes, HbA1C showed a positive correlation with creatinine and a negative one with GFR. At first, a positive correlation was observed between kidney volume and HbA1C; consequently, this group was divided into two subgroups, taking into consideration diabetes onset and duration. Consequently, this positive correlation reappeared in controlled diabetic patients for less than 20 years, while it became a negative correlation in controlled diabetic patients for 20 years and above. The correlation between the same variables was positive in uncontrolled diabetic patients (below 20 years) (r = 0.9333) and with 20 years and above with (r = 0.996), respectively. It is important to note that a larger sample size is needed for better evaluation and conclusive results, as only three uncontrolled diabetic patients for 20 years and above were enrolled in this study.

Finally, the correlation between kidney volume with GFR (r = 0.371) and HbA1C (r = −0.577) on CT scan as reported in this study is of high interest as it can be used as a valuable biomarker to easily diagnose, predict and assess CKD. For instance, an increase in creatinine by 4 mg/dL results in a decrease in kidney volume by 50 cm^3^, or an increase in GFR by 20 mL/min causes an increase in kidney volume by 20 cm^3^ approximately. On the other hand, an increase in HbA1C by 1% leads to a decrease in kidney volume by 13 cm^3^.

The current study had several limitations. First, it was conducted in one single hospital; therefore, intrinsic selection bias was present. Additionally, only right kidney measurements were taken into consideration, affecting the generalizability of the study. Moreover, renal volume measurement is not automatically segmented and depends entirely on the operator’s skills and protocol, leading to measurement and/or detection bias. Finally, there was a small sample size in the second and fourth groups.

## 5. Conclusions

In this novel study, we have reported for the first time the correlation between both GFR and HbA1C with kidney volume measured using CT scan. This study aimed to facilitate the diagnosis and quantification of kidney damage and its prognosis by physicians by simply measuring the kidney volume on CT images rather than sending the patient for ultrasound exams and correlating the ultrasound measurements with laboratory GFR and HbA1C. GFR was more powerful than creatinine when correlated with age. Kidney volume was shown to decrease faster in males than females, in addition to revealing a negative correlation between GFR and HbA1C on CT images, and thus can be used as a powerful tool to predict CKD, especially in resource-poor settings. Moreover, the positive and negative correlations of HbA1C with creatinine and GFR, respectively, provide concrete evidence that HbA1C can serve as a good biomarker in studying renal function in diabetic CKD patients. This remarkable association created a potential “link” or “formula” that can help physicians to predict variations in kidney size in newly diagnosed CKD patients, in addition to assessing treatment efficiency and outcome via laboratory testing, thus eliminating the need for repeating the CT scan.

However, further investigation should be performed, initiating standards regarding GFR, HbA1C and kidney volume correlations, shedding light on various different categories including diabetes duration and medications taken in diabetic kidney disease patients, as well as relating HbA1C results to intervals of age, different CKD stages and kidney volume.

## Figures and Tables

**Figure 1 diagnostics-11-00789-f001:**
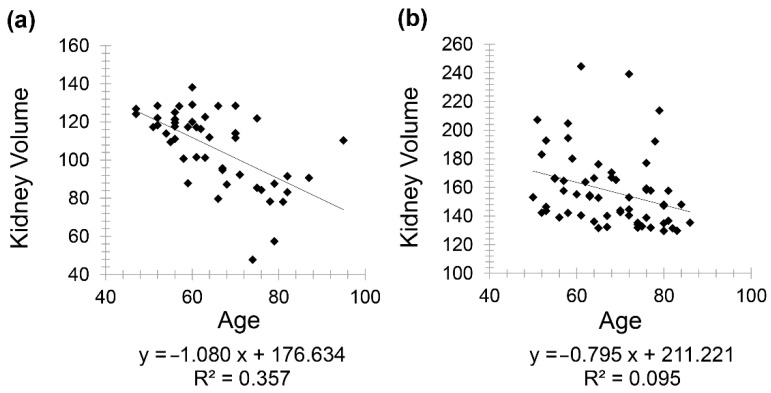
Scatter plot graphs showing negative correlation between kidney volume and age (*r* = −0.597) in males (**a**) and (*r* = −0.676) in females (**b**) in healthy patients.

**Figure 2 diagnostics-11-00789-f002:**
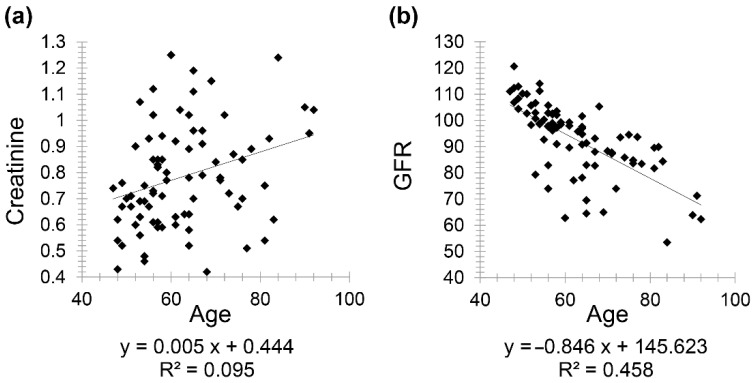
Scatter plot graphs showing (**a**) positive correlation between creatinine and age (*r* = 0.308) and **(b**) negative correlation between GFR and age (*r* = −0.676) in healthy patients.

**Figure 3 diagnostics-11-00789-f003:**
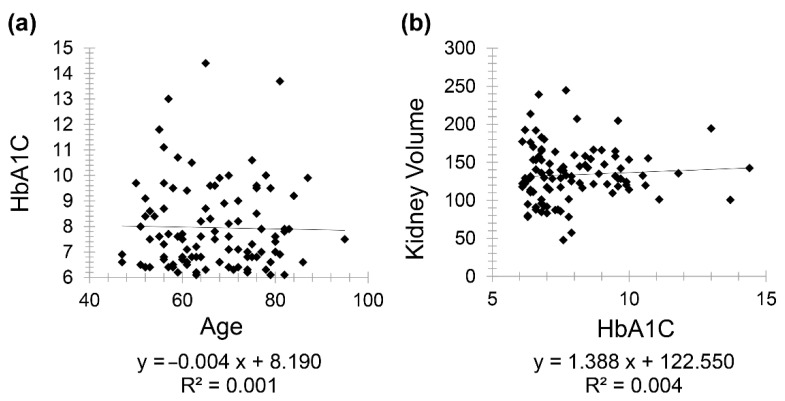
Scatter plot graphs showing (**a**) very weak negative correlation between HbA1C and age (*r* = −0.031) and (**b**) weak positive correlation between kidney volume and HbA1C (*r* = 0.063) in diabetic patients.

**Figure 4 diagnostics-11-00789-f004:**
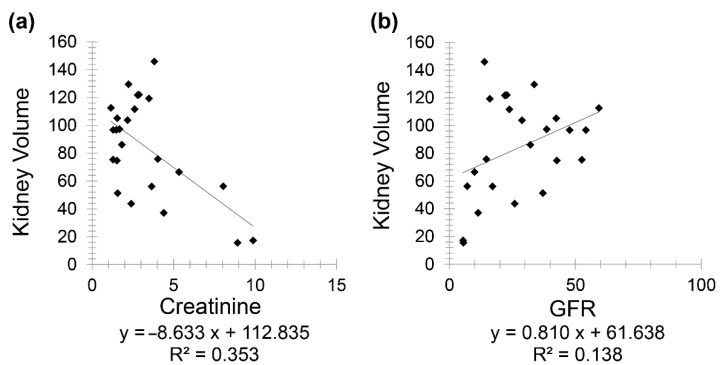
Scatter plot graphs showing (**a**) negative correlation between kidney volume and creatinine (*r* = −0.594) and (**b**) positive correlation between kidney volume and GFR (*r* = 0.371) in CKD patients.

**Figure 5 diagnostics-11-00789-f005:**
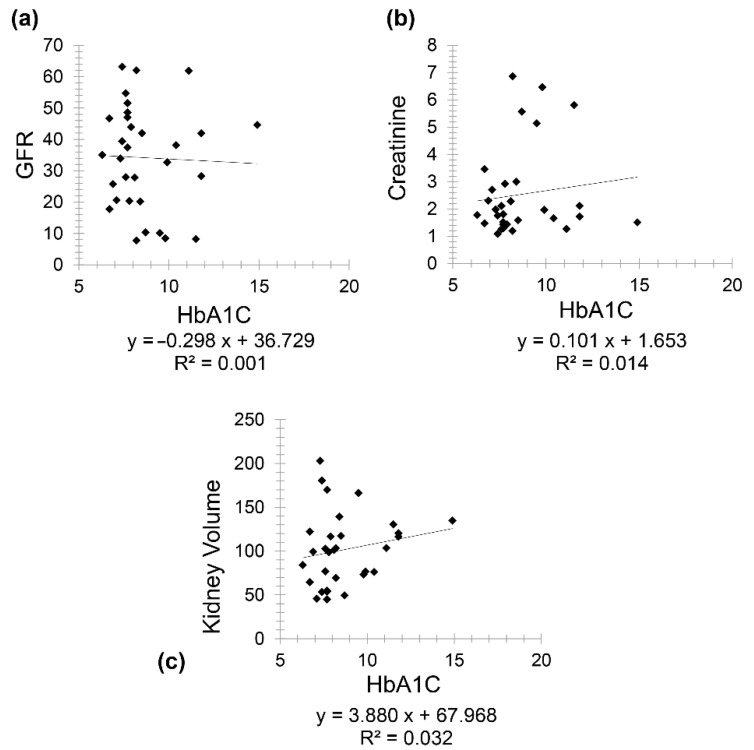
Scatter plot graphs showing (**a**) negative correlation between GFR and HbA1C (*r* = −0.031), (**b**) positive correlation between creatinine and HbA1C (*r* = 0.118), and (**c**) positive correlation between kidney volume and HbA1C (*r* = 0.178) in diabetes and CKD patients.

**Figure 6 diagnostics-11-00789-f006:**
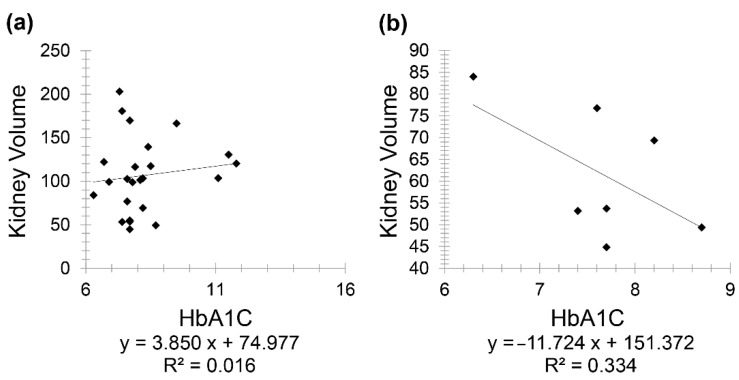
Scatter plot graphs showing (**a**) positive correlation between kidney volume and HbA1C in controlled diabetic patients (*r* = 0.126) and (**b**) stronger negative correlation (*r* = −0.577) between the same variables in controlled diabetic patients for 20 years and above.

**Figure 7 diagnostics-11-00789-f007:**
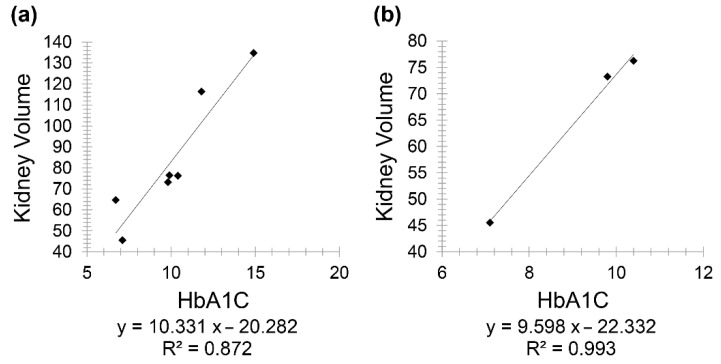
Scatter plot graphs showing (**a**) positive correlation between kidney volume and HbA1C (*r* = 0.9333) and (**b**) positive correlation (*r* = 0.996) between the same variables in uncontrolled diabetic CKD patients for 20 years and above.

## Data Availability

The data presented in this study are available on request from the corresponding author.

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
