# Peer review of "Correlation of Kidney Size on Computed Tomography with GFR, Creatinine and HbA1C for an Accurate Diagnosis of Patients with Diabetes and/or Chronic Kidney Disease"

_diagnostics, 2021, doi:10.3390/diagnostics11050789_

Round 1
Reviewer 1 Report
Please provide the IRB approval number.
The authors should precise the limitations of the study.
Reviewer 2 Report
The number of cases is small; you do not consider the congenital variations in volume and size of the kidney, also between the right and lkeft one. You do not consider previous radiological studies on kidneys volume by CT and theor limits. It is difficult to pose indication to renal biopsy only on kidney volume variations; besides albuminuria is one of the many test to detect a progressive renal failure. Your method to calculate kidney volume does not consider3-D reconstructed images.
Reviewer 3 Report
-Add the uniqueness of this study.
-Add more on the basis of this disease in the introduction
-Image analysis by both observers with an inter-observer agreement
-Discuss the role of advanced imaging such asDTI using these refs
-Abdel Razek A, Al-Adlany M, Alhadidy A, Atwa M, Abdou N. Diffusion tensor imaging of the renal cortex in diabetic patients: Correlation with urinary and serum biomarkers. Abdom Radiol 2017;42:1493-1500
-Correct English language ion through the manuscript
-Compare your results with other studies on the same issue in the discussion section.
-Discuss the merits and limitations of the technique applied
-Update of references
Round 2
Reviewer 2 Report
I accept the proposed manuscript.